# Fabrication, Performance, Characterization and Experimental Calibration of Embedded Thin-Film Sensor for Tool Cutting Force Measurement

**DOI:** 10.3390/mi13020310

**Published:** 2022-02-17

**Authors:** Yunping Cheng, Wenge Wu, Lijuan Liu, Yuntao Zhang, Zhenyu He, Ding Song

**Affiliations:** School of Mechanical Engineering, North University of China, Taiyuan 030051, China; ypchengbk@163.com (Y.C.); liulijuan@nuc.edu.cn (L.L.); zyt7262@126.com (Y.Z.); hezhenyu0229@163.com (Z.H.); sd199402@163.com (D.S.)

**Keywords:** thin-film strain sensor, cutting force, preparation process, performance characterization

## Abstract

Thin-film strain sensors are widely used because of their small volume, fast strain response and high measurement accuracy. Among them, the thin-film material and preparation process of thin-film strain sensors for force measurement are important aspects. In this paper, the preparation process parameters of the transition layer, insulating layer and Ni-Cr alloy layer in a thin-film strain sensor are analyzed and optimized, and the influence of each process parameter on the properties of the thin film are discussed. The surface microstructure of the insulating layer with Al_2_O_3_ or Si_3_N_4_ transition layers and the film without transition layer were observed by atomic force microscopy. It is analyzed that adding a transition layer between the stainless steel substrate and insulation layer can improve the adhesion and flatness of the insulation layer. The effects of process parameters on elastic modulus, nanohardness and strain sensitivity coefficient of the Ni-Cr resistance layer are discussed, and electrical parameters such as the resistance strain coefficient are analyzed and characterized. The static calibration of the thin-film strain sensor is carried out, and the relationship between the strain value and the output voltage is obtained. The results show that the thin-film strain sensor can obtain the strain generated by the cutting tool and transform it into an electrical signal with good linearity through the bridge, accurately measuring the cutting force.

## 1. Introduction

The cutting force directly affects the quality of the work-piece and tool life as one of important parameters in metal cutting processes. Accurate real-time monitoring of cutting force is helpful to study the mechanism of the cutting process and effectively control machining quality and tool life. Therefore, many researchers have done a lot of work on cutting force [1,2,3,4,5,6]. Cutting force dynamometers include strain dynamometers, piezoelectric dynamometers, current dynamometers, capacitive and hydraulic dynamometers. Among them, the strain dynamometer and piezoelectric dynamometer are the main dynamometers used. The piezoelectric dynamometer measures the cutting force through the piezoelectric effect of piezoelectric crystals [7,8]. The strain dynamometer measures the cutting force through the strain effect [9].

In recent years, with the rapid development of microelectromechanical systems (MEMS), thin-film deposition and microelectro mechanical system technology have been developed rapidly. The thin-film strain sensor is one of the kinds of dynamometer which can monitor the cutting process in real time [10,11]. The material and quality are important to the performance of strain sensors. As a result, some scholars have studied various functional layers. For the transition layer film, Zhang et al. studied a series of TiN films with different N/Ti ratios and phase evolution and mechanical properties of TiN films [12]. The hardness of the film increases with the increase of the chemometric ratio to the TiN phase, while the resistance to plastic deformation ratio of H^3^/E^2^ (H:hardness/E: elastic modulus) as an index of the toughness decreases [12]. Chen et al. deposited TiN films on 304 stainless steel surfaces by multi-arc ion plating and studied their tribology properties in a nitrogen atmosphere with GCr15 steel balls. Results showed that the TiN (200) phase has higher smoothness [13]. Chou et al. used hollow cathode discharge ion plating (HCD-IP) technology to deposit a TiN film on a 304 stainless steel surface. Hardness values ranged from 14.9 to 33.6 GPa and increased with increasing film thickness. The stacking factor of TiN film is 0.62–0.99, which increases with the increase of film thickness [14]. Qi et al. prepared nitrogen-doped titanium thin-films by direct current reactive magnetron sputtering and added different proportions of nitrogen doping into argon. Nitrogen atom energy densely arrayed hexagonal (hcp) α-Ti phase is transformed into Ti crystal (100) and Ti (002) is oriented to face-centered cubic (fcc) TiN phase. An appropriate partial pressure of nitrogen can effectively reduce the roughness and control the internal stress [15].

For insulating film, Kishore Kumar et al. used reactive RF magnetron sputtering technology to deposit Si3N4 films on surface-treated and untreated aluminum alloy substrates, observing that the lattice strain of the film decreases with the increase of grain size [16]. Nemanič et al. used reactive RF magnetron sputtering to deposit 500 and 700 nm thick amorphous Si_3_N_4_ films on an Eurofer substrate [17]. Batan et al. used reactive DC magnetron sputtering to prepare Si_3_N_4_ films on stainless steel substrates, preliminarily studied the influence of nitrogen pressure on chemical composition, microstructure and purity [18].Yao et al. deposited silicon nitride films on silicon (100) and 316L stainless steel substrates by pulse reactive closed unbalanced magnetron sputtering. They studied the influence of the N_2_ component on its chemical composition, tribology and wetting behavior [19].

Petley et al. deposited Ni-Cr alloy thin-films as the sensitive layer film at room temperature using magnetron co-sputtering technology and found that when it contained 80% Ni with 20% Cr it showed the most obvious columnar structure, had the highest resistivity, indentation hardness and elastic modulus, and good mechanical and electrical properties [20]. Lai et al. deposited Ni_80_Cr_20_ alloy films on copper foil, glass and silicon substrates, respectively, by DC magnetron sputtering. The resistivity of Ni-Cr film deposited on copper foil is higher than that of glass or silicon substrates. However, the resistance temperature coefficient (TCR) of Ni-Cr thin-films is not affected by the different substrates [21]. The deposition parameters such as sputtering power, substrate temperature and argon pressure were optimized by the Taguchi method. The polarity analysis shows that the sputtering power is the most important factor affecting the electrical properties of the nickel-chromium alloy film, and the low temperature resistance coefficient of 374.78 ppm/K is obtained [22].

Thin-film strain sensors is one of new type of microsensors which it can be integrated with tools and avoid direct contact with the workpieces, and interference without the cutting process, while allowing for data acquisition very close to the tip of the tool. This research group has done some research on the measurement of cutting force with embedded thin-film strain sensors, focusing on the measurement mechanism, structure design and preparation process of thin-film strain sensors [23,24,25,26,27,28,29].

The materials and quality of the thin-film are important to obtain good strain sensor performance. In this paper, the effect of the material and technology parameters of the membrane system film on the film quality and performance are analyzed, mechanical and electrical performance of the resistor grid layers are discussed, and the sensor strain coefficient is calibrated by experiments. As a result, we can determine the best film materials, optimize the process parameters, improve the performance of the film system and microsensor, and provide technical support for the application of strain film sensors in the measurement of cutting force.

## 2. Experimental Study of Thin-Film Strain Sensor Preparation

### 2.1. Preparation Process Flow of Thin-Film Strain Sensor

As shown in Figure 1, the preparation process of thin-film strain sensor mainly includes substrate preparation, sputtering transition layer, sputtering insulating layer, sputtering resistance grid layer, etching resistance grid structure, performance test and packaging.

The material of substrate can be selected from among AISI 1045, Ti6Al4V, AISI 304, C2800 brass and Al 1060. The mechanical properties and the macro- and micromorphology of the pretreated parts are shown in Table 1. There is small difference of tiny lines on the macroscopic surface of AISI 1045, Ti6Al4V, C2800 brass, Al 1060. There are many crisscross scratches on surface of AISI 1045 and Ti6Al4V, and scratches on surface of AISI 1045 are deeper. Scratches on surface of C2800 brass and Al 1060 are in the same direction. There are minimal scratches on surface of AISI 304. Stainless steel is often used as a substrate material due to its corrosion resistance, heat resistance and good mechanical elastic behavior [30,31,32]. Therefore, stainless steel was selected as the substrate material. Table 2 lists the physical properties and application schemes of membrane materials for thin-film strain sensors. Figure 2 shows that the Structure of thin-film strain sensor and resistance grid. When Si_3_N_4_ is deposited directly on the stainless steel substrate as an insulating layer, cracks will appear. As a result, the transition layer between the insulating layer and the substrate improves the performance of the insulating layer.

### 2.2. Preparation and Test Scheme of Membrane System of Thin-Film Strain Sensor

#### 2.2.1. Preparation Process of Si_3_N_4_ Film

During the preparation of insulating layer Si_3_N_4_ film, sputtering power, sputtering pressure, gas flow ratio and substrate temperature are the four main deposition process parameters. In the preliminary test, several different values are selected for each process parameter by single factor experiments, and four levels for every process parameter are selected. In order to optimize the process parameters, the four levels of each factor are studied by orthogonal experiments. The level values of the orthogonal factors of the four process parameters are shown in Table 3. The orthogonal experimental design and test results of the four deposition process parameters are shown in Table 4.

#### 2.2.2. Preparation Process of Al_2_O_3_ and TiN Transition Layer Films

Stainless steel 304 is used as the substrate material because of its corrosion resistance, heat resistance and good mechanical elasticity [30,31,32]. Cracks would appear and spread and eventually the coating could fall off when Si_3_N_4_ film is directly deposited. In order to increase the adhesion between Si_3_N_4_ and the substrate, a transition layer need to be added. In this paper, Al_2_O_3_ and TiN are used as transition layer materials. The thin-film strain sensor can meet the requirements of high strain rate and improve the measurement accuracy and service life of the sensor.

Before depositing the transition layer, the stainless-steel substrate was subjected to surface treatment to obtain a surface roughness of 30–50 nm, which was ultrasonically cleaned with acetone and absolute alcohol for 20 min. sputtering power, sputtering pressure, and Ar:O_2_ flow ratio are the main process parameters in the preparation process of Al_2_O_3_ film. In this experiment, the target spacing is always 30 mm. The change range of sputtering power is 60~100 W, sputtering pressure is 0.5~2.5 Pa, Ar:O_2_ flow ratio is 10:1~50:1. The Ar:N_2_ flow ratio of nitrogen and argon and the substrate negative bias are the main process parameters in preparing process of TiN film. The range of Ar:N_2_ flow ratio is 20:1~60:1, and substrate negative bias is 0~100 V.

#### 2.2.3. The Preparation Process of Ni-Cr Alloy Film

Sputtering power, sputtering pressure, substrate negative bias and substrate temperature are the main process parameters in the preparation process of Ni-Cr alloy films. The relationship between process parameters and the deposition rate of nickel-chromium films is analyzed by orthogonal experiment. The factor level table of the four process parameters is shown in Table 5. The change range of sputtering power is 80~140 W, 20 W intervals between each level, sputtering pressure is 1.0~2.5 Pa, substrate negative bias is 0–150 V, and substrate temperature is 20–300 °C.

Table 6 shows the orthogonal experiment design and test results of deposition rate of Ni-Cr alloy film.

The Ni-Cr alloy film is etched into the shape of resistance grid by ion beam etching. After sputtering the transition layer, insulating layer and Ni-Cr film on the substrate, the Ni-Cr alloy film resistance grid is obtained by ion beam etching through photolithography and development. Incident angle, Argon flow and substrate negative bias are main process parameters of ion beam etching. The factor level table of the three process parameters is shown in Table 7.

Table 8 shows the orthogonal experiment design and test results of etching rate, surface roughness and resistivity.

### 2.3. Result Analysis

#### 2.3.1. Variance Analysis of Orthogonal Test of Process Parameters for Si_3_N_4_ Film

Table 9 is the result of variance analysis according to the orthogonal test results of Table 4 which the orthogonal experimental design and test results of the four deposition process parameters of Si_3_N_4_ film.

Comparing the F value of the mean deviation square sum and the mean deviation squares errors of each parameter, the influence degree of the process parameters on deposition rate of the Si_3_N_4_ film can be obtained as follows: Sputtering power A > Gas flow ratio C > Sputtering pressure B > Substrate temperature D.

It can be seen from Table 10 that the influence degree of the process parameters on surface roughness of the Si_3_N_4_ film can be obtained as follows: Sputtering power A > Gas flow ratio C > Sputtering pressure B > Substrate temperature D.

#### 2.3.2. Analysis of Process Parameters for Al_2_O_3_ Film

From Figure 3, we can observe the change trend of alumina film deposition rate with the change of process parameters. From Figure 3a–e, the range of deposition rate is 0.9~7 (nm/min) when the Ar:O_2_ flow ratio and sputtering power are constant. The deposition rate of the film increases first and then decreases with the sputtering pressure gradual increasing. The range of deposition rate is 2~13.3 (nm/min) when the Ar:O_2_ flow ratio and the sputtering pressure are held constant. The deposition rate of the film increases with the increase of power. The range of deposition rate is 19.1–33.2 (nm/min) when the power and sputtering pressure are constant. The deposition rate of the films increases with the increase of Ar:O_2_ flow ratio. This shows that Ar:O_2_ flow ratio has the greatest influence and the sputtering pressure has the least influence on the deposition rate.

#### 2.3.3. Analysis of Process Parameters for TiN Films

The deposition rate and surface roughness values of the prepared TiN film are shown in Table 11 and Table 12.

From Table 11, it can be seen that the deposition rate decreases as the Ar:N_2_ flow ratio decreases. The decrease rate changes from 66.1 to 21.9 with the Ar:N_2_ flow ratio changes from 60:1 to 20:1 when the substrate negative bias is 100 V. The decrease rate always change from big to small with the Ar:N_2_ flow ratio changing from 60:1 to 20:1 under the same substrate negative bias. It can be seen that the deposition rate decrease with the reduction of total gas.

The decrease rate change from 66.1 to 56.4 with the substrate negative bias changing from 100 to 0 when the Ar:N_2_ flow ratio is 60:1. The decrease rate increases first and then decreases with the substrate negative bias changing from 100 to 0 when the Ar:N_2_ flow ratio is 30:1 or 25:1 or 20:1. When the total gas reduces, the influence of negative bias on the deposition rate increases, then the deposition rate increases with the decrease of substrate negative bias and the deposition rate would decrease when the substrate negative bias disappears. That is, the deposition rate would decrease when the substrate negative bias is large. As a result, it can be seen that the effect of Ar:N_2_ flow ratio on deposition rate is greater than substrate negative bias.

From Table 12, the substrate negative bias increases from 0 V to 100 V, the maximum difference of surface roughness is 20 (Ar:N_2_ flow ratio is 60:1) and the minimum difference is 12 (Ar:N_2_ flow ratio is 20:1) when Ar:N_2_ flow ratio is constant. It can be seen that the surface quality is good when Ar:N_2_ flow ratio is smaller. When the substrate negative bias of the substrate is constant, the difference of surface roughness is the range of 13–20 nm, and the surface roughness decreases with the decrease of Ar:N_2_ flow ratio, but the surface roughness value is irregular when the substrate negative bias is 100 V. It can be seen that in order to obtain better surface quality, a lower substrate negative bias and gas flow ratio are required.

#### 2.3.4. Analysis of Orthogonal Test of Process Parameters for Ni-Cr Film

The data of Table 6 are collected with range analysis, where the range values of the effects of sputtering power, sputtering pressure, negative bias and substrate temperature on deposition rate are 18.62, 0.71, 5.62 and 0.50, respectively. As a result, the influence degree of the process parameters on deposition rate can be obtained as follows: sputtering power > substrate negative bias > sputtering pressure > substrate temperature.

It can be seen from Table 6 that the deposition rate increases gradually with the increase of sputtering power and substrate negative bias. The velocity of the sputtering particles bombarding the substrate increases with the increase of sputtering power, which increases the deposition rate of the film. However, with the increase of the deposition rate, particle energy would increase. As a result, particle energy would affect the surface flatness of films. As the strain sensitive layer in the sensor, the roughness and flatness of the Ni-Cr film are very important for the theoretical calculation of resistance and the accurate measurement of strain. Therefore, the flatness of the film should be ensured while the deposition rate increases.

#### 2.3.5. Analysis of Orthogonal Test of Etching Process Parameters for Ni-Cr Film

The influence of various factors on the film etching rate and the film surface roughness is discussed with the range analysis method according to the results of the orthogonal experimental. As shown in Table 13, where *K_i_* (*i* = 1, 2, 3) is the average of measurement results at each level, and *R_j_* (j = A, B, C) is the range of each factor, and where X is incident angle, Y is argon flow, Z is substrate negative bias.

Figure 4 shows the relationship of etching rate, roughness and parameters.

Figure 4a is the range analysis of the process parameters on the film etching rate, the degree of influence: substrate negative bias > argon flow > incident angle. In the etch process, the substrate negative bias and arc voltage are superimposed to form ion source energy. As the energy of the ion beam increases, the energy of the particles transferred to the substrate increases and the material removal speed accelerates and the etching rate increases. Figure 4b shows the range analysis of the process parameters on the film roughness, and the degree of their influence: incident angle > substrate negative bias > argon flow. The Ni-Cr alloy film roughness is important for the protective layer on it, because it is related to the adhesion between the two films. The roughness value is smaller when the incidence angle is 45°.

## 3. Performance Characterizations of Sensors

### 3.1. Characterization of Microscopic Morphology and Mechanical Properties of Thin-Films

#### 3.1.1. 3-D Morphology of Aluminum Oxide and TiN Transition Layer

Figure 5 shows the 3-D morphology of silicon nitride film, silicon nitride film containing alumina transition layer and silicon nitride film containing titanium nitride transition layer. It can be seen that the average surface roughness of silicon nitride film without transition layer is between 18–33 nm; the average roughness surface of silicon nitride film with alumina film as transition layer is between 17–25 nm, the number of peaks and valleys on significantly less, the surface morphology is relatively smooth; the average roughness surface of silicon nitride film with titanium nitride film as transition layer is between 11–12 nm, the number of peaks and valleys on significantly least, the surface morphology is more smooth. Insulation film quality is especially important for the manufacture of resistance grid layer.

#### 3.1.2. Mechanical Properties of Ni-Cr alloy Resistance Grid Layer

Ni-Cr alloy film has the largest elastic modulus, nano-hardness and resistivity. Aluminum oxide and silicon nitride thin-films were sputtered on 304 stainless steel substrates before sputtering the Ni-Cr alloy films. Then the Ni-Cr alloy films were deposited on the surface of the silicon nitride thin-films. The data are recorded in Figure 6, respectively.

From Figure 6, the elastic modulus Ni-Cr alloy film first increases and then decreases with the increase of sputtering pressure, and then stabilizes. It increases with the increase of sputtering power and substrate negative bias. Therefore, in order to increase the elastic modulus of Ni Cr alloy film, the sputtering power and substrate negative bias can be appropriately increased. The elastic modulus Ni-Cr alloy film has small influence on of when the substrate temperature is below 200 °C. The value of the elastic modulus increases from 1F81 to 198 when the temperature is from 200 °C to 300 °C. It can be seen that nano hardness of the film are not affected by the change of sputtering pressure, and decrease with sputtering power and substrate temperature. Nanohardness changes periodically with substrate negative bias. The nanohardness mainly changes in the range of 8.3–8.6 GPa.

The strain sensitivity coefficient of Ni-Cr alloy film is the most important index to measure the mechanical properties of the film sensor. The larger the coefficient value, the better the sensitivity of the film sensor. Figure 7 shows the values of the strain sensitivity coefficient varying with the elastic modulus. The values of the strain sensitivity coefficients are 1.78, 1.76, 1.74 and 1.56 with elastic modulus of 176, 180, 185 and 198 GPa, respectively. The coefficient value decreases with the increase of elastic modulus.

Combined with the conclusion of Figure 6, it is necessary for elastic modulus to be a lower value in order to obtain a higher sensitivity coefficient (≧ 1.7). Since the strain sensitivity coefficient of 1.78 corresponds to the elastic modulus of 176 GPa, the corresponding process parameters can be selected according to the requirements of elastic modulus. For example, the sputtering pressure is between 2–2.6 Pa, the sputtering power is less than 110 W, the negative bias voltage of the substrate is less than 20 V, and the substrate temperature is less than 200 °C.

### 3.2. Characterization of Electrical Properties of Ni-Cr Films with Metal Sensitive Layers

#### 3.2.1. Effect of Geometric Structure Parameters of Resistance Gate on Resistance Strain Coefficient

Through the calculation of the geometric parameters of the resistance grid, the theoretical resistance value of resistance grid is 1200 Ω. The effect of thin-film sensor with different lengths, width and thickness (as shown in Figure 2) on the resistance and resistance strain coefficients is studied. The dimensions are shown in Table 14.

The resistance strain coefficients of the prepared thin-film sensor are analyzed by strain measurement, and the load range of 0–400 N is selected for the stretch test. Figure 8 shows the results of the stretch test of the thin-film sensor with resistance grid length, width and thickness under the condition of single factor. It can be seen that the width of the resistance grid has a relatively large influence on the resistance strain coefficient, of which the width of the 0.1 mm has the largest effect. When the strain is less than 300 μɛ, the resistance strain coefficients of different lengths or thicknesses are almost the same. When the strain exceeds 300 μɛ, the resistance strain coefficients of different lengths or thicknesses are slightly different, but there is little difference between the values. The results show that the structure size of the resistance grid has little effect on the resistance strain coefficient.

#### 3.2.2. Effect of Etching Parameters on Resistivity

The range analysis is performed for the film etching resistivity test results in Table 13. Figure 9 is the parameter range analysis of surface roughness, it can be seen the degree of influence: incidence angle > substrate negative bias > argon flow. With the increase of incidence angle, the film resistance first increases and then decreases, and the inflection point appears at 45°. It can be seen that the resistivity is the lowest at 45°.In order to obtain high resistivity, the influencing order and level factors of process parameters can be adjusted as needed.

## 4. Calibration of Sensor Strain Coefficient

In the experiment of calibrating, unidirectional force is applied to the thin-film strain sensor by tensile tester. There is experiment device in Figure 10a, using a DH5929 dynamic signal test and analysis system to collect strain values of 16 thin film resistor grids that form four groups of Wheatstone bridges, as shown in Figure 10b. The experimental data are shown in Figure 10d, the curve of strain value and output voltage, getting the relationship as Equation (1):(1)$$U_{out}=\frac{1}{2}k_{n}\varepsilon_{n}U_{in}$$
where *U_out_* is the output voltage of the circuit, *k_n_* is the strain sensitivity coefficient of the thin-film resistance gate, *ε_n_* is the strain value of the thin-film resistance gate, and *U_in_* is the input voltage of the circuit.

The arrangement of 16 resistance grids is shown in Figure 10b. There are four resistor grids arranged on the two resistor grids I and II, as shown in Figure 10c. Resistance grids R_1_–R_8_ adopts strain gauge I, which is 45 degrees from the horizontal. On the strain gauge I, resistance grids R_1_, R_3_, R_5_ and R_7_ adopt the grid 1 which is the longitudinal arrangement, R_2_, R_4_, R_6_ and R_8_ adopt the grid 2 which is the horizontal arrangement. Resistance grids R_9_–R_16_ adopt the grid 3 and 4 on the strain gauge II, in which resistance grids 3 and 4 are the longitudinal arrangement. It can be seen from Figure 10d that good linearity exists between strain and electrical signal, which is conducive to the accurate measurement of cutting force. In the Wheatstone bridge circuit, the curve slope between the output voltage and the thin film sensor is 0.5 kn. The equation in Figure 10d, the resistance strain coefficients k_α_, k_β_, k_γ_ and k_δ_ of the thin-film resistance grid are 1.88, 1.88, 1.71 and 0.26 respectively. The resistance strain coefficients kα and kβ are equal, because resistance grids R_9_–R_16_ are longitudinal arrangement. Resistance grids R_9_–R_16_ are in the same stress state during the tensile test. Resistance strain coefficients k_γ_ is 1.71 less than 1.88 because resistance grid R_1_, R_3_, R_5_ and R_7_ are the longitudinal arrangement and 45 degrees from the horizontal. It is in accordance with the resistance strain coefficients of Ni-Cr film between 1~2. Resistance strain coefficients *k_δ_* is 0.26 because resistance grid R_2_, R_4_, R_6_ and R_8_ are the horizontal arrangement and 45 degrees from the horizontal. In order to obtain the maximum strain, the length direction of the resistance grid needs to be consistent with the load direction, thus obtain a larger output voltage and achieve the purpose of measuring the cutting force.

## 5. Conclusions

### 5.1. Effect of Process Parameters on Each Film

The influence degree of the process parameters on the deposition rate of the Si_3_N_4_ insulating layer is followed by sputtering power, gas flow ratio, substrate temperature, and sputtering pressure. The Ar:O_2_ flow ratio has more influence on the deposition rate of Al_2_O_3_ film than sputtering power and sputtering pressure. The influence degree of the parameters of the deposition rate of the TiN transition layer is followed by Ar:N_2_ flow ratio, and substrate negative bias. The substrate negative bias has a greater effect on the surface roughness than Ar:N_2_ flow ratio.

The influence degree of process parameters on the deposition rate of Ni-Cr alloy film is followed by sputtering power, substrate negative bias, sputtering pressure, substrate temperature. The influence degree of the etching process parameters on the etching rate of Ni-Cr alloy film is followed by the substrate negative bias, argon flow, and the incident angle, and the order of the influence on the surface roughness is the incident angle, substrate negative bias, and the argon flow. As a result, the film with good performance can be obtained by adjusting the parameters that have a great impact on the film performance.

### 5.2. Effect of Process Parameters on Each Film

Al_2_O_3_ and TiN as the transition layer between 304 stainless steel substrate and Si_3_N_4_ insulating layer can reduce the protrusion on the film surface and improve the surface flatness and uniformity of the Si_3_N_4_ insulating layer. As a result, it is necessary to increase the transition layer between 304 stainless steel substrate and Si_3_N_4_ insulating layer in order to improve the performance of insulating layer

### 5.3. Effect of Process Parameters on Each Film

For the Ni-Cr alloy film, the elastic modulus of the Ni-Cr alloy film is inversely proportional to sputtering power, directly proportional to substrate negative bias and sputtering pressure. When the substrate temperature is less than 200 °C, the elastic modulus of the film has little effect. The Nano-hardness of the films is inversely proportional to the sputtering pressure and substrate temperature, and varies periodically under the substrate negative bias of 0–160 V.

### 5.4. Effect of Process Parameters on Each Film

Greater strain can be obtained when the length direction of the resistance grid consistent with the load direction, thus obtain a larger output voltage and achieve the purpose of measuring the cutting force. Thin film strain sensors can be selectively placed according to different cutting conditions.

## Figures and Tables

**Figure 1 micromachines-13-00310-f001:**
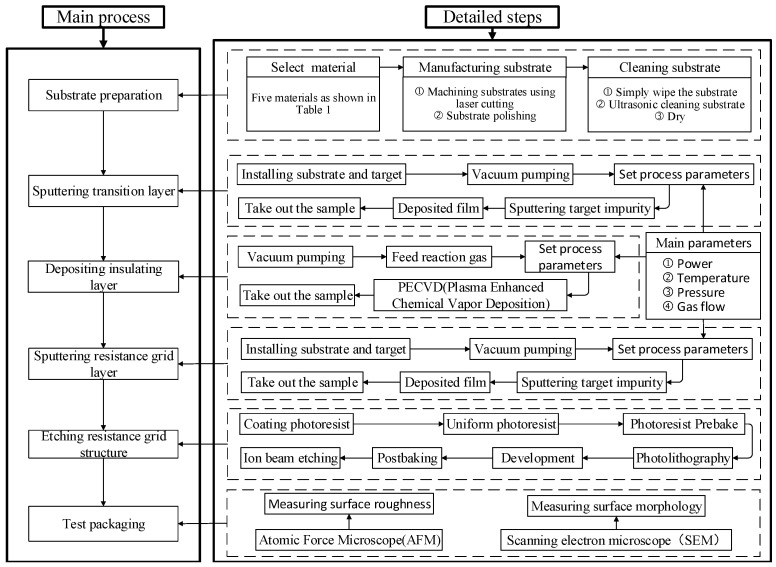
Process flow diagram.

**Figure 2 micromachines-13-00310-f002:**
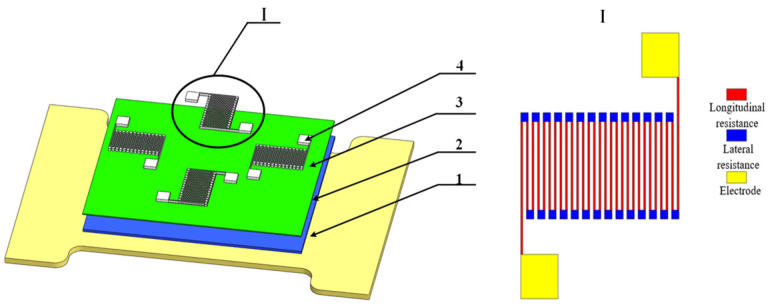
Structure of thin-film strain sensor and resistance grid. 1, Elastic substrate; 2, Transition layer; 3, Insulating layer; 4, Resistance grid layer.

**Figure 3 micromachines-13-00310-f003:**
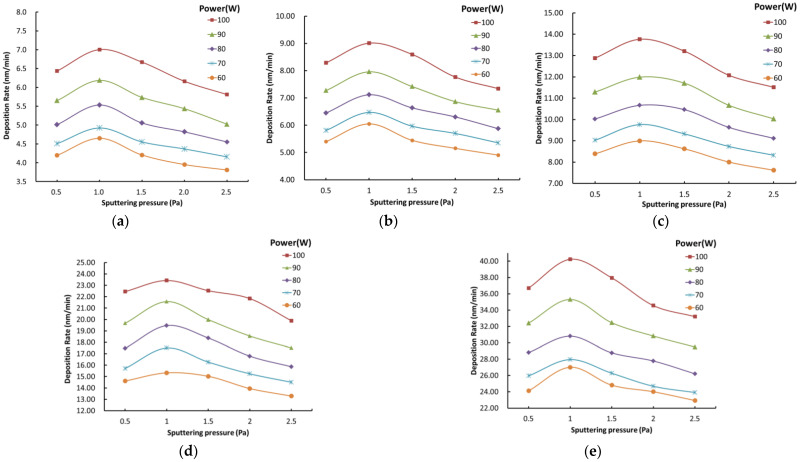
The influence of process parameters on the deposition rate of Al_2_O_3_ film. (**a**) Ar:O_2_ = 10:1; (**b**) Ar:O_2_ = 20:1; (**c**) Ar:O_2_ = 30:1; (**d**) Ar:O_2_ = 40:1; (**e**) Ar:O_2_ = 50:1.

**Figure 4 micromachines-13-00310-f004:**
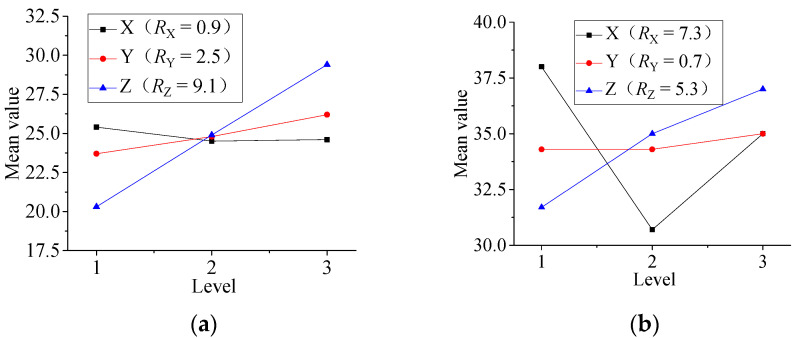
The relationship of etching rate, roughness and parameters. (**a**) Parameter range analysis of film etching rate; (**b**) Parameter range analysis of film roughness.

**Figure 5 micromachines-13-00310-f005:**
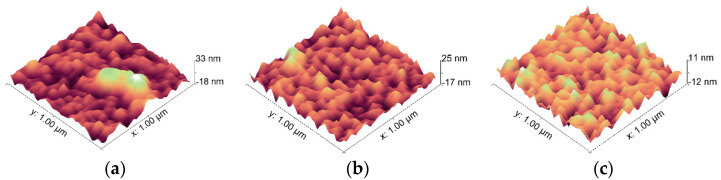
3-D morphology of silicon nitride thin-film by atomic force microscopy (MFP-3D). (**a**) Silicon nitride thin-film; (**b**) Silicon nitride film with Aluminum oxide transition layer; (**c**) Silicon nitride film with TiN transition layer.

**Figure 6 micromachines-13-00310-f006:**
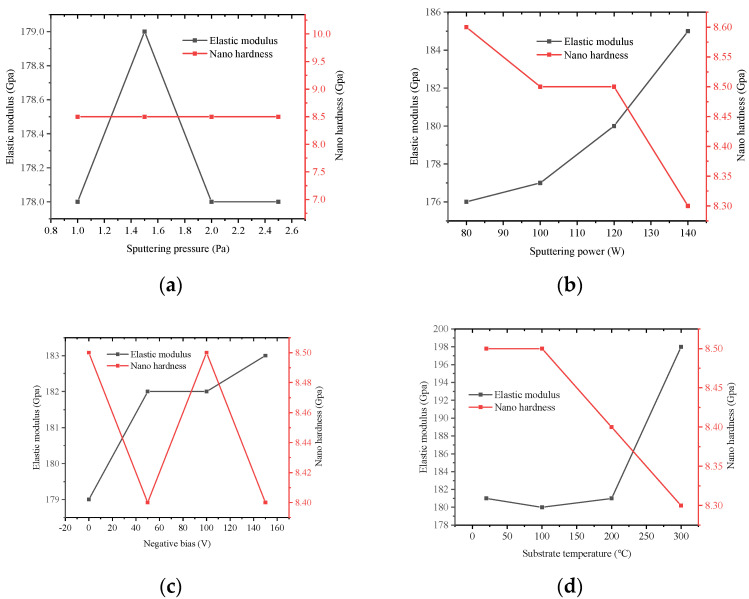
Effect of different process parameters on elastic modulus and nano-hardness. (**a**) Sputtering pressure curve; (**b**) Sputtering power curve; (**c**) Substrate negative bias curve; (**d**) Substrate temperature curve.

**Figure 7 micromachines-13-00310-f007:**
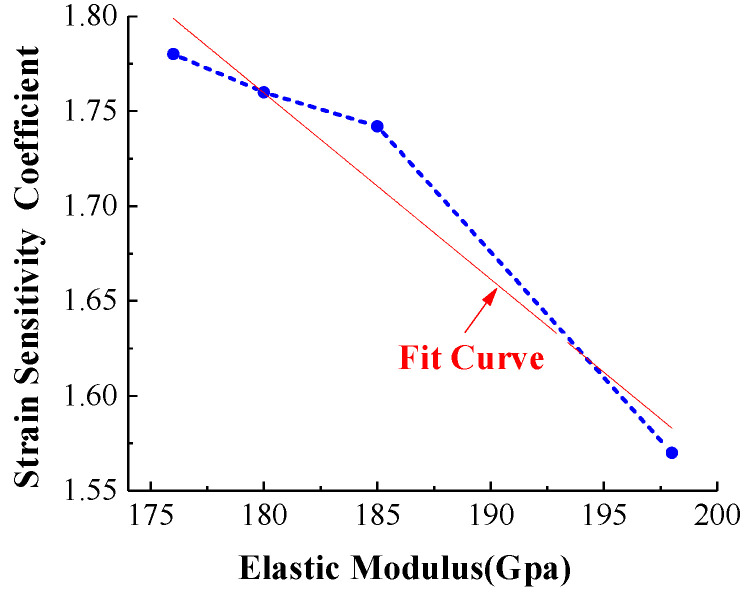
Test results of strain sensitivity coefficient.

**Figure 8 micromachines-13-00310-f008:**
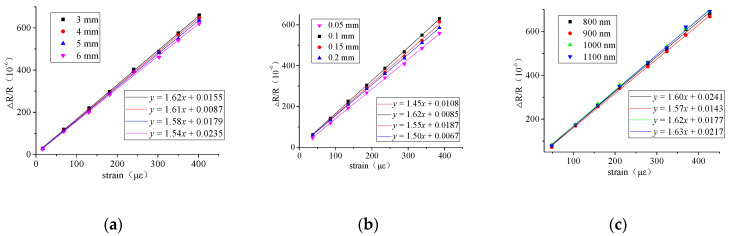
Tensile test results of structure parameters of resistor gratings prepared by etching method. (**a**) Effect of resistance grid length on resistance strain coefficient; (**b**) Effect of resistance grid width on resistance strain coefficient; (**c**) Effect of resistance grid thickness on resistance strain coefficient.

**Figure 9 micromachines-13-00310-f009:**
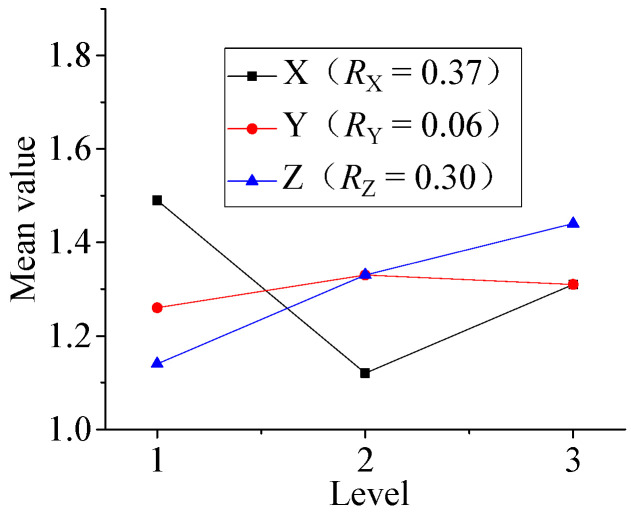
Parameter range analysis of film resistivity.

**Figure 10 micromachines-13-00310-f010:**
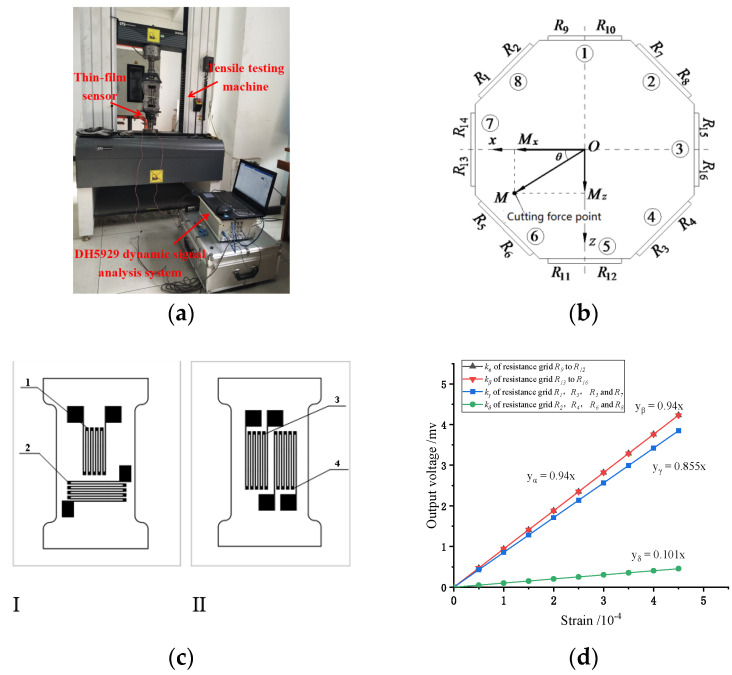
Tensile test of thin film strain sensor. (**a**) stretching device; (**b**) Distribution diagram of resistance grid; (**c**) Resistor grids I and II; (**d**) Output voltage and strain diagram of thin-film sensor.

**Table 1 micromachines-13-00310-t001:** Substrate material characteristics, macroscopic features and microscopic features.

Material	AISI 1045	Ti6Al4V	AISI 304	C2800 Brass	Al 1060
Elastic modulus (GPa)	210	113	200	100	69
Shear modulus (GPa)	79	44	75	37	26
Thermal Expansion Coefficient (10^−6^/K)	11.59	7.89	15	18	23.6
Yield strength (MPa)	355	825–895	205	239	135
Poisson’s ratio	0.31	0.34	0.29	0.33	0.33
Material appearance	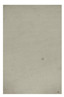	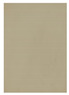	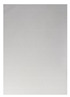	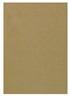	
Ultra depth of field micrograph (500 times)	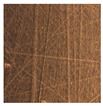	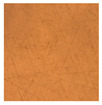	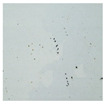	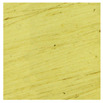	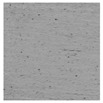

**Table 2 micromachines-13-00310-t002:** Material properties and application of various types of films.

	Al_2_O_3_ Film	TiN Film	Si_3_N_4_ Film	Ni_80_Cr_20_ Film
Elastic Modulus (GPa)	357	279	304	186
Poisson’s Ratio	0.21–0.27	0.25	0.24	0.38
Thermal Expansion Coefficient (10^−6^/K)	7–10.3	7.4	2.45	14
Nano-hardness (GPa)	14.2	33.6	17.6	9.7
Application	Transition layer	Transition layer	Insulating layer	Resistance grid layer

**Table 3 micromachines-13-00310-t003:** Level table of orthogonal test factors of Si_3_N_4_ film.

Level	A	B	C	D
	Sputtering Power (W)	Sputtering Pressure (Pa)	Gas Flow Ratio	Substrate Temperature (°C)
1	60	0.6	0	ambient temperature
2	80	0.8	0.125	100
3	100	1.0	0.2	200
4	120	1.2	0.25	300

**Table 4 micromachines-13-00310-t004:** L_16_(4^4^) Orthogonal test design and results of Si_3_N_4_ film.

Number	A	B	C	D	Deposition Rate *v*/(nm/min)	Surface Roughness *Ra/*(nm)
1	60	0.6	0	20	5.442	48
2	60	0.8	0.125	100	3.080	47
3	60	1.0	0.2	200	3.091	48
4	60	1.2	0.25	300	3.245	49
5	80	0.6	0.125	200	4.031	52
6	80	0.8	0	300	6.052	56
7	80	1.0	0.25	20	3.175	49
8	80	1.2	0.2	100	4.678	51
9	100	0.6	0.2	300	4.579	52
10	100	0.8	0.25	200	5.012	49
11	100	1.0	0	100	8.546	54
12	100	1.2	0.125	20	6.149	45
13	120	0.6	0.25	100	7.125	52
14	120	0.8	0.2	20	8.328	54
15	120	1.0	0.125	300	9.058	56
16	120	1.2	0	200	13.012	59

**Table 5 micromachines-13-00310-t005:** Level table of orthogonal test factors of Ni-Cr alloy film.

Level	A	B	C	D
	Sputtering Power (W)	Sputtering Pressure (Pa)	Substrate Negative Bias (V)	Substrate Temperature (°C)
1	80	1.0	0	20
2	100	1.5	50	100
3	120	2.0	100	200
4	140	2.5	150	300

**Table 6 micromachines-13-00310-t006:** L_16_(4^4^) Orthogonal test design and results of Ni-Cr alloy film.

Number	A	B	C	D	Deposition Rate (W)
1	80	1.0	0	20	47.67
2	80	1.5	50	100	49.11
3	80	2.0	100	200	50.89
4	80	2.5	150	300	52.77
5	100	1.0	50	200	56.21
6	100	1.5	0	300	54.26
7	100	2.0	150	20	61.06
8	100	2.5	100	100	58.55
9	120	1.0	100	300	63.85
10	120	1.5	150	200	66.98
11	120	2.0	0	100	61.41
12	120	2.5	50	20	62.69
13	140	1.0	150	100	71.03
14	140	1.5	100	20	69.66
15	140	2.0	50	300	68.21
16	140	2.5	0	200	66.01

**Table 7 micromachines-13-00310-t007:** Level table of orthogonal test factors of ion beam etching.

Level	X	Y	Z
	Incident Angle (°)	Argon Flow (m^3^/s)	Substrate Negative Bias (V)
1	20	1 × 10^–5^	350
2	45	1.25 × 10^–5^	450
3	70	1.5 × 10^–5^	550

**Table 8 micromachines-13-00310-t008:** Ion beam etching orthogonal test design and results.

Number	Incident Angle (°)	Argon Flow (m^3^/s)	Substrate Negative Bias (V)	Etching Rate *v*/(nm/min)		Surface Roughness Ra/(nm)	Resistivity *Ρ* (μΩ·m)
				Ni-Cr	Photoresist AZ6140		
1	20	1 × 10^–5^	350	19.8	18.2	35	1.21
2	20	1.25 × 10^–5^	450	25.0	23.4	38	1.57
3	20	1.5 × 10^–5^	550	31.5	27.8	41	1.68
4	45	1 × 10^–5^	450	23.6	22.6	31	1.10
5	45	1.25 × 10^–5^	550	29.1	26.3	33	1.19
6	45	1.5 × 10^–5^	350	20.9	18.1	28	1.06
7	70	1 × 10^–5^	550	27.6	26.6	37	1.46
8	70	1.25 × 10^–5^	350	20.2	17.4	32	1.16
9	70	1.5 × 10^–5^	450	26.1	24.2	36	1.32

**Table 9 micromachines-13-00310-t009:** Variance analysis of film deposition rate test results of Si_3_N_4_ film.

	Degree of Freedom	Deviation Square Sum	Mean Deviation Squares Sum	F Ratio
A: Sputtering power/(W)	3	75.706	25.235	142.571
B: Sputtering pressure/(Pa)	3	4.837	1.612	9.107
C: Gas flow ratio	3	31.234	10.411	58.819
D: Substrate temperature/(°C)	3	0.778	0.259	1.463
Error	3	0.53	0.177	

**Table 10 micromachines-13-00310-t010:** Variance analysis of film surface roughness Ra test results of Si_3_N_4_ film.

	Degree of Freedom	Deviation Square Sum	Mean Deviation Squares Sum	F Ratio
A: Sputtering power/(W)	3	114.688	38.229	12.481
B: Sputtering pressure/(Pa)	3	1.688	0.563	0.184
C: Gas flow ratio	3	51.188	17.063	5.571
D: Substrate temperature/(°C)	3	38.688	12.896	4.21
Error	3	9.19	3.063	

**Table 11 micromachines-13-00310-t011:** Effect of process parameters on film deposition rate of TiN film.

Deposition Rate (nm/min)							
		Negative Bias (V)					
		100	80	60	40	20	0
**Ar:N_2_**	60:1	66.1	61.1	60.2	59.7	58.4	56.4
	40:1	46.9	42.3	41.8	40.7	39.5	38.6
	30:1	33	34.3	35.6	37.4	35.7	33.5
	25:1	25.7	29.7	31.3	35.3	34.5	31.3
	20:1	21.9	25.6	28.5	31.3	32.8	29.3

**Table 12 micromachines-13-00310-t012:** Effect of process parameters on surface roughness Ra of TiN film.

Surface Roughness (nm)							
		Negative Bias (V)					
		100	80	60	40	20	0
**Ar:N_2_**	60:1	27	31	35	39	43	47
	40:1	25	27	31	33	39	43
	30:1	39	34	27	29	31	35
	25:1	31	27	19	21	24	30
	20:1	29	21	15	20	22	27

**Table 13 micromachines-13-00310-t013:** Range analysis of thin-film etching experiment results.

	Film Etching Rate *v/*(nm/min)			Film Roughness Ra/(nm)			Resistivity ρ (μΩ·m)		
	X	Y	Z	X	Y	Z	X	Y	Z
*K* _1_	25.4	23.7	20.3	38.0	34.3	31.7	1.49	1.26	1.14
*K* _2_	24.5	24.8	24.9	30.7	34.3	35.0	1.12	1.33	1.33
*K* _3_	24.6	26.2	29.4	35.0	35.0	37.0	1.31	1.31	1.44
*R_j_*	0.9	2.5	9.1	7.3	0.7	5.3	0.37	0.06	0.30
	Influence level Z > Y > X			Influence level X > Z > Y			Influence level X > Z > Y		

**Table 14 micromachines-13-00310-t014:** Geometrical dimensions of thin-film sensor with different length of resistance grid.

Sample	Longitudinal Resistance Grid Length	Resistance Value: 1200 Ω; Longitudinal Resistance Grid Width: 0.1 mm; Longitudinal Resistance Grid Thickness: 800 nm			
		Lateral Resistance	Longitudinal Resistance	Single Resistance Grid Resistance Value (Ω)	Electrode
1	3 mm	0.4 × 0.4 mm	28	41.25	2 × 2 mm
2	4 mm		21	55	
3	5 mm		17	68.75	
4	6 mm		14	82.5	

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
