# Peer review of "Fabrication, Performance, Characterization and Experimental Calibration of Embedded Thin-Film Sensor for Tool Cutting Force Measurement"

_micromachines, 2022, doi:10.3390/mi13020310_

Round 1

Reviewer 1 Report

This work is pertinent to the development of a force sensor for the purpose of measuring cutting force signals. It is an interesting study which can be considered for publication after several modifications are carried out:

A part of the abstract section from "The material of substrate ... Ni-Cr resistor grid" need to be rephrased as the sentences and not related between each others and several mistakes are noted. 

In the Introduction section, abbreviations such as "H" or "E" need to be explained. "Tri-bological" should be written without the hyphen. Every reference must be mentioned by the name of the first author with "et al." if the number of authors exceeds 3, otherwise all authors should be mentioned. Large paragraphs should be split into smaller. Finally, at least 10-15 more references need to be added e.g. regarding previous works on measuring cutting forces with developed sensors and the novelty of the current study needed to be stressed more clearly.

In Table 1, why is "material appearance" important to be mentioned? Gpa or Mpa should be corrected to GPa and MPa respectively. In page 4 "gird" should be replaced with "grid". Regarding the values chosen for the various orthogonal experiments, how were these values obtained? Appropriate references should be added to justify these values. The units for Argon flow should be checked again and expressed in SI units if possible.

In section 2, at pages 6-9 the authors number some phrases which are not subsection titles e.g. "1. Table 9 is the result of variance ...of Table 4". In order to avoid confusion, the numbering of these phrases should be omitted. Subsection format should be checked in order to be in accordance with journal format.

Why did the authors not perform some machining tests in order to validate their sensor? Which are the requirements to be considered for the measurement of cutting forces with this type of sensors? 

Author Response

Dear Professor

There are three documents in total, one is the reply to the review comments, one is the word of the modified paper, and another is the PDF of the modified paper. Please check it.

Reviewer 2 Report

The authors prepared an article named “Fabrication, Performance, Characterization and Experimental Calibration of Embedded Thin-Film Sensor for Tool Cutting Force Measurement” which looks good and carries important findings about tool cutting force measurement, however the manuscript need to be improved according to following comments:

The abbreviations should be explained before they are used.

I also propose to add a nomenclature to the article.

Please demonstrate in the abstract novelty of the study. Also show practical significance.

At first, the article has 12 references which cannot be accepted for such kind of topic in this esteemed journal. Considering the vast amount of paper in the field, the reviewer asks to add many citations to this manuscript.

The authors need to explain why they used orthogonal arrays.

Last paragraph of the introduction should simply explain the main aim of the work.

It should be widened the related studies about this field in the introduction. There is need to discuss the importance of the cutting forces in machining. Also, I highly propose to address the different types of cutting force measurements namely, piezoelectric, strain-gauge, etc. It is important to add the following articles and further in this direction:

“Design, development and testing of a four-component milling dynamometer for the measurement of cutting force and torque”

“Three-component, strain gage based milling dynamometer design and manufacturing”

In the subtopic of NiCr film etching process, the text before the tables should be improved.

Please add more explanations for all figures and tables. They are inadequate.

The results and discussion parts have to be improved referring updated papers and analyzing in detail.

It is also important to add citations including cutting force measurements in Mdpi journals. Some of them are listed below, however, the authors need to add more.

“Modeling of cutting parameters and tool geometry for multi-criteria optimization of surface roughness and vibration via response surface methodology in turning of AISI 5140 steel”

“A review of indirect tool condition monitoring systems and decision-making methods in turning: critical analysis and trends”

“Optimization and analysis of surface roughness flank wear and 5 different sensorial data via tool condition monitoring system in turning of AISI 5140”

“Parametric optimization for cutting forces and material removal rate in the turning of AISI 5140”

Author Response

Dear Professor:

The reply to the review comments is on the end of paper. Please check it!

Best regards,

Cheng Yunping,

Round 2

Reviewer 1 Report

The authors have performed most of the required modifications. Thus, the manuscript can be recommended for publication. It is suggested that the authors perform two minor modifications before the paper is published:

In line 48, "hard" should be corrected to "hardness". The units for Ar flow should be in appropriate units (perhaps in SI),  not "SIm", e.g. m^3/s or something more appropriate.

Reviewer 2 Report

I have reviewed the paper and see that the authors have been made many contributions to article. I think the paper is ready for publishing at the final version. My decision is about to accept it.